# Bringing Data Analytics to the Design of Optimized Diagnostic Networks in Low- and Middle-Income Countries: Process, Terms and Definitions

**DOI:** 10.3390/diagnostics11010022

**Published:** 2020-12-24

**Authors:** Kameko Nichols, Sarah J. Girdwood, Andrew Inglis, Pascale Ondoa, Karla Therese L. Sy, Mariet Benade, Aloysius Bingi Tusiime, Kekeletso Kao, Sergio Carmona, Heidi Albert, Brooke E. Nichols

**Affiliations:** 1FIND, 1202 Geneva, Switzerland; kameko@thenicholsgroupllc.com (K.N.); kekeletso.kao@finddx.org (K.K.); Sergio.carmona@finddx.org (S.C.); 2Health Economics and Epidemiology Research Office, Department of Internal Medicine, School of Clinical Medicine, Faculty of Health Sciences, University of the Witwatersrand, Johannesburg 2193, South Africa; 3USAID Global Health Supply Chain Programme, Procurement and Supply Management, International Business Machines, Arlington, VA 22202, USA; Andrew.inglis@ibm.com; 4African Society for Laboratory Medicine, Addis Ababa 5487, Ethiopia; POndoa@aslm.org; 5Amsterdam Institute for Global Health and Development, 1105 BP Amsterdam, The Netherlands; 6Department of Global Health, Amsterdam University Medical Center, 1105 AZ Amsterdam, The Netherlands; 7Department of Global Health, Boston University School of Public Health, Boston, MA 02118, USA; rsy@bu.edu (K.T.L.S.); mbenade@bu.edu (M.B.); 8Department of Epidemiology, Boston University School of Public Health, Boston, MA 02118, USA; 9USAID Global Health Supply Chain Programme, Procurement and Supply Management, Chemonics International, Arlington, VA 22202, USA; ABingiTusiime@ghsc-psm.org; 10FIND, Cape Town 7925, South Africa; heidi.albert@finddx.org; 11Department of Medical Microbiology, Amsterdam University Medical Center, 1105 AZ Amsterdam, The Netherlands

**Keywords:** diagnostic network optimization, data analytics, low- and middle-income countries

## Abstract

Diagnostics services are an essential component of healthcare systems, advancing universal health coverage and ensuring global health security, but are often unavailable or under-resourced in low- and middle-income (LMIC) countries. Typically, diagnostics are delivered at various tiers of the laboratory network based on population needs, and resource and infrastructure constraints. A diagnostic network additionally incorporates screening and includes point-of-care testing that may occur outside of a laboratory in the community and clinic settings; it also emphasizes the importance of supportive network elements, including specimen referral systems, as being critical for the functioning of the diagnostic network. To date, design and planning of diagnostic networks in LMICs has largely been driven by infectious diseases such as TB and HIV, relying on manual methods and expert consensus, with a limited application of data analytics. Recently, there have been efforts to improve diagnostic network planning, including diagnostic network optimization (DNO). The DNO process involves the collection, mapping, and spatial analysis of baseline data; selection and development of scenarios to model and optimize; and lastly, implementing changes and measuring impact. This review outlines the goals of DNO and steps in the process, and provides clarity on commonly used terms.

## 1. Diagnostics: An Essential Health Systems Component

“Without diagnostics, medicine is blind.” [1] Diagnostics are an essential component of healthcare systems and are integral to many clinical decisions in confirming disease; monitoring treatment; recognizing complications such as drug resistance; preventing the spread of disease and antimicrobial resistance; and enhancing surveillance for early disease detection and monitoring [2]. Furthermore, the essential role of diagnostics in advancing universal health coverage (UHC) is acknowledged in the 2nd List of Essential In Vitro Diagnostics, and the role of diagnostics to ensure global health security is outlined in the International Health Regulations (IHR) [3,4]. Yet diagnostics are often unavailable, inaccessible, or too costly for patients who need them in low- and middle-income countries (LMICs), and frequently remain overlooked and under-resourced [5,6]. Moreover, where testing capacity is available, it is often underutilized and of variable quality, and result delivery is inconsistent or too lengthy to render it clinically relevant [7].

There are many important considerations for the role of diagnostics in disease detection, such as determining the appropriate setting for a test, whether the tests will be used appropriately, whether the clinical utility of the test is proven in a particular population, and whether the test results will be available in time to inform patient-care decisions or public-health measures. A quick and accurate diagnosis can ensure that the right care and treatment is administered to the patient in time, allowing for a more effective treatment and monitoring. In the case of infectious diseases, including human immunodeficiency virus (HIV) and tuberculosis (TB), patients who remain undiagnosed can also unknowingly transmit infection to others and develop permanent sequelae. Evidence-based optimization of diagnostic services can close the gap on access to diagnostics and bring us closer to achieving UHC and IHR.

## 2. The Importance of Diagnostic Networks

As countries move towards UHC, they should ensure that diagnostic testing is affordable, of high quality and that there is equity in access [8]. One way to ensure availability of diagnostic services to all patients regardless of where they first seek care within the health system would be to place diagnostic testing at every health facility. There are, however, multiple reasons why this is not feasible or advisable. Firstly, not all diagnostic tests are suitable for use at point-of-care or at peripheral health facilities due to skilled human-resource constraints and infrastructure requirements. Secondly, managing and maintaining quality becomes difficult when more testing sites are added and testing is decentralized. This is especially the case when low numbers of tests may be conducted at individual health facilities, driving up the cost per test. Further, a significant financial investment would be required to establish and maintain the required infrastructure and equipment. In the majority of LMICs these budgets are severely constrained and have other competing priorities.

Instead, to address these challenges, diagnostic services are made available through an integrated, tiered national **laboratory network** where the various tiers are determined by their diagnostic test menus based on population testing needs, infrastructure requirements, and resource constraints [8,9]. The purpose of this tiered national laboratory network is to provide integrated diagnostic services for clinical and public health systems [8]. Next, specimen referral systems link patients and/or samples at peripheral health facilities to laboratories or testing sites within the network, such that service offerings are broadened beyond what is offered onsite and tests are performed at the most appropriate tier [10,11,12,13].

A **diagnostics network** is the interconnected system that is used to yield a diagnosis either within a clinical or public-health setting. Although the terms “diagnostic network” and “laboratory network” are often used interchangeably, we distinguish between the two, with a diagnostic network going beyond the definition of the laboratory network to encompass a more patient-centered, coordinated network where tests are accessible, accurate and adaptable, and results are produced quickly [14]. A diagnostic network includes rapid diagnostic and point-of-care testing that may occur outside of a laboratory in the community and clinic settings; it also elevates the supportive role of the specimen referral system to a critical role for the functioning of the diagnostic network. An illustrative example of a tiered national diagnostic network is found in Figure 1 below.

The design and planning of diagnostic networks to date has relied mostly on manual methods and expert consensus, with a limited application of data analytics. However, this approach is not ideal to analyze complex and multivariate datasets, including demand for services, locations and capacity, as well as exploring the current baseline state and future potential scenarios under a range of applied constraints, including costs, allowable service distances, and the turnaround time of results. More recently, there have been numerous efforts to improve this methodology, and thus the term “**diagnostic network optimization**” (DNO) and related terms are used frequently. When these terms are used, they are used with a wide range of definitions. As such, it is useful to define terms used around this type of network analysis. Furthermore, it is important to also understand what questions we can answer with DNO, and what it can do to strengthen diagnosis and national health systems in the context of UHC and the requirements of IHR. The terms are introduced and explained throughout the text, and then each term is defined in more detail in Table 1.

## 3. Diagnostic Network Optimization

One way to inform improvement of the broader diagnostic network for a country is to conduct diagnostic network design and optimization. DNO matches testing demand and capacity (e.g., through device placement and test-demand referral) to increase access, improve efficiency, and develop routing for the specimen referral network [7]. It is an analytical process that selects the best network configuration from available alternatives based on objectives (minimizing costs, maximizing access, minimizing turnaround time) and variables (transport, device placement, etc.) within given constraints; in other words, a mathematical calculation solving for the optimal combination of variables with the goal of achieving the objectives [20]. **Diagnostic network design** can help highlight the tradeoffs that exist between the different objectives and constraints and why it is important to be clear about the objectives of the DNO upfront during the design process. For example, while it might make sense to place point-of-care HIV viral-load devices at every health facility in order to maximize population coverage, it would not be the optimal solution if the system is constrained by costs. Alternatively, while it might not be cost-effective to place point-of-care HIV viral-load devices at low-volume, hard-to-reach facilities, if the model is maximizing access (or **equity** in access), these competing dimensions would need to be balanced. The design process helps reveal these tradeoffs between efficiency and effectiveness.

To date, the majority of DNO work in LMICs has been focused on the infectious disease space. This is driven in part by the investment in viral-load monitoring for patients on antiretroviral treatment for HIV infection, early infant diagnosis for infants born to HIV-positive mothers, and TB diagnosis [7,20,21,22,23,24]. Given the siloed nature of funding, the vast majority of DNO exercises have been **vertical** in nature or disease-specific, likely leading to inefficiencies. Instead of siloed DNOs, **integration** across a full basket of **essential diagnostics** is possible and is especially critical when integration across test devices and specimen-referral systems results in efficiencies. The basket of essential diagnostics will differ from country to country, but should satisfy the priority healthcare needs of the population and be selected based on disease prevalence, public health relevance, and evidence of utility, accuracy, and comparative cost-effectiveness [3]. The selection of the essential diagnostics is not explicitly covered in network optimization.

Although a computer model or software is not required for a simple spatial analysis, for our purposes we are limiting network design and optimization to sophisticated processes that are done with the aid of computer software that enables integration of Geographic Information System (GIS) data, health-service capacity, and diagnostic demand to prioritize and weigh trade-offs among the various priorities and objectives. While a network optimization approach has been used in improving supply chains in the corporate sector, application to optimizing the design of diagnostic networks and specimen referral is relatively new in LMICs [25]. Some examples of the use of DNO in the field include the use of mathematical modeling linked with GIS software in Zambia and South Africa, as well as the use of specific software (Llamasoft’s Supply Chain Guru) in Lesotho, Kenya, and the Philippines [20,21,22,26,27,28,29,30,31]. The focus of DNO in Zambia has been on the optimization of viral-load access and has seen the number of viral tests performed double within a year [21,22,27,32]. DNO exercises in South Africa have optimized the diagnostic network for CD4 testing, placement of viral-load point-of-care, and also optimized the specimen routing for multiple diagnostics [26,33,34]. In Lesotho, Kenya, and the Philippines, DNO has informed instrument placement and specimen referral within the TB diagnostic network [20,28,29,30]. The need for an accessible, easy-to-use software focused on diagnostic networks was identified following work in several countries using proprietary tools, and development and piloting of such a tool (OptiDx) is now underway [35,36,37].

The DNO process encompasses the following activities: geographic mapping and baseline model creation, scenario creation and analysis, network optimization, and evaluation.

### 3.1. Geographic Mapping and Creation of the Current State Network Model

The first step in the optimization process is the collection of baseline data and geographic **mapping** of the current diagnostic network. Location data for facilities (**collection points**, **testing facilities**, and **hubs**) can be visualized on a geographic map and overlaid with additional diagnostic network data, including: (1) demand for the respective diagnostic; (2) capacity of testing (based on human resources and equipment); and (3) referral linkages (which collection point refers specimens to which testing facility through which hub(s), if necessary). Once the data are mapped, analysis of the spatial relationship between populations (demand), collection points, and testing facilities, as well as device utilization and cost allows identification of gaps in the current diagnostic network in terms of demand, capacity, and **coverage**, and opportunities for improvement that can be explored in alternative scenarios. This analysis of the diagnostic network exposes the different trade-offs that decision-makers might need to consider.

### 3.2. Scenario Development

Once the current network state is mapped and spatially analyzed, alternative **scenarios** are created to test out various future state configurations for the diagnostic network [20]. These scenarios should be discussed with key stakeholders, reflect stakeholder priorities, and address key questions that can be evaluated through **scenario analysis**, for example: “*What proportion of HIV viral-load results can be returned within an agreed-upon turnaround time?*” or “*How will device capacity and placement need to evolve to meet projected TB testing demand?*” or “*What is the optimal mix of devices to fill remaining capacity gaps?*” or “*In serving demand and turnaround time for people living with HIV, how can we ensure demand for other indications is not negatively impacted?*” Once a subset of prioritized scenarios is decided upon, a model will be created for each scenario. The use of scenario analysis aims to provide a representation of the defined scenarios and their impact on the diagnostic network. After the models are created for each scenario, network optimization can begin. A set of outputs, common across all scenarios, can then be compared.

### 3.3. Measuring Success of Diagnostic Network Optimization

It is important to identify how diagnostic network design and optimization might improve the network and to choose indicators that measure this potential improvement. The analytical processes of design and optimization aim to improve the **effectiveness**, **efficiency,** and **adaptability** of the diagnostic network [7,14]. These three primary objectives are important to define for diagnostic network analysis and can be evaluated pre- and post-optimization:Effectiveness: An effective diagnostic network should ensure that essential diagnostics are available, accessible, return results within clinically relevant turnaround times, and are of high quality. These are intermediate outcomes; final health outcomes in terms of the impact of a diagnostic network on public health and disease surveillance are not measured here.

Are essential diagnostics **available** to those who need them? This is measured by whether the tests are available geographically onsite, or via a referral system, and if so, if there is sufficient capacity. It can be measured as the number of health facilities that offer a given diagnostic onsite (or via a referral system) divided by the total number of health facilities in a country, weighted by facility volume. Integration of testing on a platform might increase the availability of certain diagnostic tests.

Are essential diagnostic services **accessible** to those who need them? Accessibility can be measured as the number of people who receive a given diagnostic divided by the total number of people estimated to require a given diagnostic across a specified time period.

Are essential diagnostic results returned in a timely manner? Access can be further qualified to include **turnaround time**—the proportion of diagnostic services that is accessible within a prescribed turnaround time period. Turnaround time can be measured as the amount of time taken between specimen collection to return of results at the requesting facility.

Are the diagnostic services available of high **quality**? Quality of the diagnostic services is measured based on the probability that the test result is reproducible and accurate. This is a function of good manufacturing practice, trained and competent staff following standard testing procedures, low specimen-rejection rates, and internal and external quality assurance.

Efficiency: Efficiency of the diagnostic network can be measured in several ways: (1) cost outcomes: the ratio of financial resources consumed (testing costs incurred) and the output (number of tests completed) to determine a cost per test completed, and/or the cost per correct test result returned to the patient; (2) Device utilization: how efficiently a device is operated, measured as the average number of tests conducted on a piece of laboratory equipment divided by the maximum number of tests that can be conducted on that piece of laboratory equipment across a specified time period. Higher device utilization and testing integration frequently result in a lower cost per test [21]. The focus is on overall efficiency of the entire network and not individual devices.Adaptability: Adaptability measures the ability of a diagnostic network to meet current testing demands (effectiveness) and to adjust to changing needs within the diagnostic network, whether as a result of the addition of a disease program or specimen type, introduction of new technology, or a disease outbreak [14]. Both the Ebola outbreak of 2014/2015 and the current SARS-CoV-2 pandemic have highlighted the need to ensure that robust integrated systems and platforms are in place to enable an effective public health response. Adaptability is more difficult to measure as it includes factors not reflected in the diagnostic network (for example, leadership). A proxy for adaptability would be to assess the extent to which volumes of currently supported diagnostic tests were impacted by the changing needs within the diagnostic network (e.g., disease outbreak) or how quickly new technology can be adopted and successfully deployed within the network. DNO is a key process to be used to stress-test the diagnostic network to determine how well it can withstand shocks (for example, where underutilized devices can be relocated, or where spare capacity or integrated testing can be leveraged), or how best to adopt new technology.

## 4. Conclusions

Diagnostic network design and planning have, to date, predominantly used manual methods, with limited application of data analytics. Use of a DNO approach that can integrate multiple data inputs and constraints holds promise to improve network design, aimed at enhancing the efficiency, effectiveness, and adaptability of diagnostic networks, thereby improving public health and disease surveillance. This review aimed to clarify the terms and process commonly used in DNO. Using uniform language will contribute to establishing an evidence base on the effectiveness of DNO in the future.

## Figures and Tables

**Figure 1 diagnostics-11-00022-f001:**
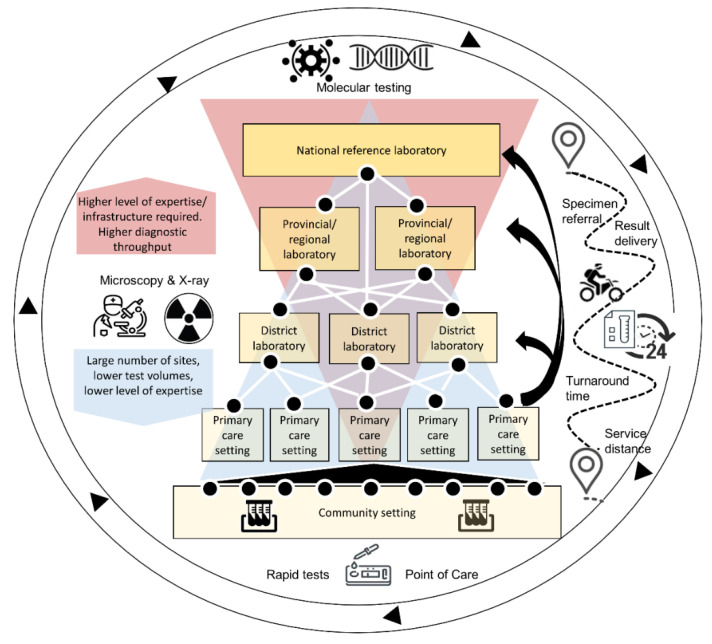
Example of a national, interconnected diagnostic network: At the higher levels of the network there are fewer sites, higher required levels of expertise, and high-throughput diagnostics such as polymerase chain reaction tests for viral detection. At the most basic levels of the network (e.g., clinic and community settings), this includes a large number of sites and rapid diagnostic tests and point-of-care; all functions are supported by a specimen referral system that ensures timely specimen referral and result delivery for all sites, regardless of service distance.

**Table 1 diagnostics-11-00022-t001:** Terms and definitions related to diagnostic network design and optimization. Terms highlighted in green address, “What is a diagnostic network and how does it compare to a laboratory network?”; items in blue address, “How can we improve the diagnostic network through analysis?”; and terms in yellow address, “What are the objectives of a diagnostic network? How do we know if it is performing well?”.

Term	Specific Definition as Related to Diagnostics Network Optimization
**Tiered laboratory network**	An integrated system of laboratories organized in tiers aligned with the public health delivery system of the country. The tiers are determined by their test menus and functions, and a **specimen referral network** ensures tests are performed at the most appropriate level of the tiered system.
**Diagnostic network**	“Diagnostic” and “laboratory” are often used interchangeably with “network” but “diagnostic network” includes all testing sites and instruments within a laboratory network as well as testing sites and instruments that fall outside a laboratory setting—e.g., rapid diagnostic- and point-of-care tests, which can be delivered in community and clinic settings. It allows for inclusion of non-laboratory testing that is part of diagnostics process, i.e., screening, radiology, etc. The role of the specimen referral network is elevated.
**Specimen referral network**	An interconnected group of specimen referral systems, which comprise all components and processes required for patient specimens to be tested at a location that differs from where the specimen was collected.
**Diagnostic network design**	Uses location data, testing demand and capacity and referral linkage data on **collection points** and **testing facilities** to build a model of the **current state/baseline** of the network and then using that to help identify gaps and opportunities and improve the network to achieve desired objectives i.e., improved access or reduced cost.
**Diagnostic network optimization (DNO)**	Optimization maximizes or minimizes an objective, by changing the variables under control subject to certain constraints. In the context of diagnostic networks, it is a computerized analytical process that designs a single diagnostic network based on objectives (minimizing costs, maximizing access, minimizing turnaround time) and variables (transport, device placement) within given **constraints** (capacity, costs); the best setting of variables that meets the objectives. Excluded from the definition are other analyses or interventions aimed at strengthening laboratory systems that do not incorporate these aspects.
**Constraints**	Limits placed on variables in the process of optimization. In DNO, constraints could include capping the total number of instruments, number of testing sites, actual equipment capacity, total costs, or ability to refer samples across administrative boundaries within a country.
**Route optimization**	Route optimization is the process of determining the most efficient route. It is more complex than simply finding the shortest path between two points. It needs to include all relevant factors such as the number and location of all the required stops on the route. This is why route optimization is mostly performed by computer algorithms that can quickly narrow down the options. Route optimization software can quickly test multiple ‘what-if’ scenarios to help fleets review the costs of different route options and resource availability, but within a given set of constraints. This is also known as vehicle route optimization.
**Collection points**	The physical location where specimens are collected from an individual requiring a diagnostic test. These may also be referred to as referring facilities or spokes.
**Hubs**	The physical location where specimens may be pooled after leaving the collection point. The hub may offer certain testing onsite or may only serve as an intermediate pooling and processing point (i.e., for centrifuging) for specimens before they reach the testing facility. Hubs may also offer quality checks and documentation points for specimens.
**Testing facilities**	The physical location where the specimens are processed. If the diagnostic test is offered onsite, then this could be the same physical location as the collection site. If testing occurs offsite, then this would be the physical location of the laboratory.
**Baseline or Current state**	Current status of the diagnostic network, including current levels of accessibility, turnaround times, interconnectedness, collection points, and testing facilities. It is a fixed point of reference that is used for comparison purposes.
**Mapping**	Refers to the geospatial mapping of the location data for collection points and testing facilities supplemented by other diagnostic network data, namely testing demand, testing capacity and referral linkages between locations; however, importantly, mapping is not synonymous with “network optimization.”
**Scenarios**	Scenarios are potential changes that can be made to the baseline or current state and are created through a subjective exercise to discuss “What if” (in terms of inputs or variables such as capacity or demand) and decide which to explore; this process is informed by the baseline situation (including mapping, spatial analysis, costs).
**Scenario analysis**	The process or technique of testing potential changes to the system to consider the potential outcomes and implications of a change to assist or improve decision making. This answers the question of “what if” and allows for comparison with the baseline or current state.
**Vertical systems or programs**	A vertical system or program only focuses on one disease or one area. For example, a vertical specimen referral system would only refer specimens for one disease program (such as for HIV, or for TB). The primary advantage of a vertical system is that the needs of an individual disease program is prioritized. The primary disadvantage is potential inefficiencies introduced by having multiple vertical systems serving one disease each [15].
**Integrated systems or programs**	An integrated system or program focuses on multiple diseases or areas. Integration can happen at multiple levels of the health system or in different areas. For example, a specimen referral system that serves primary health facilities and connects them to district level may carry more than one specimen-type for more than one disease area (this will also depend on the co-location of pickup and delivery points). Integration must be done to ensure the goals and needs for all programs are met and optimization for one program does not negatively impact other programs. Another type of integration is for testing (see **testing integration**).
**Testing integration**	Testing integration or multiplexing uses the same technology (also known as polyvalent testing platforms or multianalyte analyzers) for several assays and/or across diseases. It can lead to more efficient and cost-effective testing services. Further, diagnostic integration can help to simplify and streamline other systems, such as specimen referral, human resources and quality assurance [16].
**Coverage**	Health service coverage is defined as the extent to which target populations receive health interventions. For diagnostic services coverage, this relates to geographic coverage as well as targeted coverage of key populations such as people living with HIV (PLHIV), children, miners, other vulnerable populations, etc [17].
**Availability**	Availability is a component of physical access—in order for a service to be accessible, it must first be available. Availability is centered around testing (which requires capacity and capability) and referral systems (the tests available either onsite or via a referral system) [18].
**Access**	The focus of ‘access’ for DNO is on physical accessibility of diagnostics, and what proportion of individuals can access a diagnostic if required within a given health system.
**Turnaround time**	The time elapsed between collection of a specimen from a client and return of the results to the facility or client, and, in some cases, to the time of clinical intervention based on that result. If these data are unavailable, other intermediary turnaround times are used, e.g., testing turnaround time. In DNO, turnaround time refers to the time that the specimen is picked up to the time that the result is returned to the requesting facility [19].
**Efficiency**	Efficiency concerns the relationship between resource inputs (e.g., costs) and intermediate or final health outcomes. There are two areas of efficiency for DNO: (1) cost outcomes—the ratio of financial resources consumed (total cost of the diagnostic network) to the health outcome (the valued health system output that is created by the cost input—for example, number of correct tests results returned within a time period, etc.)) and (2) equipment utilization. Efficiency must be balanced with access and other effectiveness measures such as turnaround time.
**Maximum equipment testing capacity**	Maximum capacity, or theoretical capacity, does not take the actual testing environment into consideration—it is the manufacturer’s calculation of the instrument capacity. This may be used as the denominator in the **utilization** calculation.
**Actual equipment testing capacity**	Actual available equipment capacity takes into consideration human resource availability and capacity, site conditions, and thus it is usually less than theoretical maximum capacity. It may be used as the denominator in the **utilization** calculation.
**Utilization**	The level of usage of equipment within a set time window compared to the **maximum theoretical device testing capacity** and/or **actual available device testing capacity** within the same period, e.g., actual number of tests conducted as a proportion of the total number of tests that could have been conducted on a particular device.
**Equity**	Equity in health refers to fairness in the distribution of healthcare resources and outcomes amongst population groups defined socially, economically, demographically or geographically. As it relates to DNO, equity is a measure of the distribution and fairness of geographical access. Equity considerations are a competing dimension to efficiency considerations which can be explicitly examined through DNO. For example, whilst it might not be cost-effective to provide diagnostic access to certain hard-to-reach populations, it might be important if decision-makers value equity.
**Quality**	This refers to the quality of the diagnostic services in terms of the probability that the test result is accurate and reproducible and is a broader concept compared to external quality assessment. Quality starts with selecting and procuring the right tests produced under good manufacturing practice, validated as appropriate for the population and settings of intended use. Next, it is a function of the specimen type and collection, transport, and whether testing is done by trained and competent staff, as well as the level of quality-assurance (both internal and external, including proficiency testing) during all steps of the process.
**Adaptability**	Adaptability measures the ability of a diagnostic network to adjust to changing needs within the diagnostic network, whether as a result of an addition of a disease program, new technology, specimen type or disease outbreak.

## Data Availability

No new data was created or analyzed in this study. Data sharing is not applicable to this article.

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
