# Peer review of "Bringing Data Analytics to the Design of Optimized Diagnostic Networks in Low- and Middle-Income Countries: Process, Terms and Definitions"

_diagnostics, 2020, doi:10.3390/diagnostics11010022_

Round 1
Reviewer 1 Report
- No data regarding the diagnostics in low and middle-income countries was discussed.
- The authors generalized the diagnostics but randomly mentioned HIV and TB. A focused approach on a specific disease diagnostics will add more value to the manuscript.
- The definitions of various terms used in diagnostic optimization networks are more generalized.
- What is the significance of defining the terms?
Author Response
Reviewer 1:
- No data regarding the diagnostics in low and middle-income countries was discussed.
Response: In line 56-line 61 we discuss the current state of access to diagnostics in LMICs. In this section, we discuss how diagnostics are often unavailable, inaccessible, or too costly in LMICs. In order to illustrate this point, we have cited two seminal papers – Petti et al and Nkengasong et al.
- The authors generalized the diagnostics but randomly mentioned HIV and TB. A focused approach on a specific disease diagnostics will add more value to the manuscript.
Response: Thank you for this comment. DNO is disease-agnostic and therefore we wanted to generalize across all diagnostics. In fact, disease-specific DNO can result in inefficiencies and therefore integration across a full basket of essential diagnostics is possible and important from an efficiency point of view. However, to date, DNO exercises have been driven by infectious diseases such as TB and HIV. I would like to refer the author to the paragraph starting at line 139 where we acknowledge that the majority of work in the DNO space has focused on HIV/TB largely due to investments and siloed funding structures.
- The definitions of various terms used in diagnostic optimization networks are more generalized.
Response: Thank you for this comment. We agree with the reviewer; however, the purpose of this Review is to provide context and define DNO terms as used in this specific field. We have therefore tried to define the terms within the context of DNO in LMICs, with the goal that when these terms are referred to in the field, that all researchers and implementers work from the same set of definitions.
- What is the significance of defining the terms?
Response: We argue that it is important to clarify the terms and processes commonly used in DNO. Using uniform language will help to the establishment of an evidence-base in which robust comparisons can be made between studies and on the effectiveness of DNO in the future.
Reviewer 2 Report
The paper proposes a review/survey about diagnostic network optimization (DNO), with a description of steps in the process, and provides clarity on commonly used terms.
First section 1. must be an introduction to the paper (Introduction section). In this section, the author could describe shortly the introduction of DNO and show the main contribution of the revision proposed. Try to show if is there some paper with similar proposes.
Figure 1 is interesting and could be better described in order to relate to DNO.
Table 1 indicate and describe the terms of DNO. This table needs to give support using references.
The paper, as the title indicates, focuses on low- and middle-income (LMIC) countries, however, the paper does not bring any work or example of a set of countries that the DNO is starting to be used. This is the core of the paper, then needs better an in deep discussion.
Also, It is important to bring a short review of how rich countries work with this, such as a short comparative.
At text "Although a computer model or software is not required for a simple spatial analysis, for our purposes, we are limiting network design and optimization to sophisticated processes that are done with the aid of computer software that enables integration of Geographic Information System (GIS) data, health service capacity and diagnostic demand to prioritize and weigh trade-offs between the various priorities and objectives". The author provides a paper review, but in this sentence, the model is presented as a proposal of the author. Give to clear if is a review paper or a new model proposed.
In general, the paper brings an initial discussion about DNO in LMIC countries.
Author Response
Reviewer 2:
- The paper proposes a review/survey about diagnostic network optimization (DNO), with a description of steps in the process, and provides clarity on commonly used terms.
- First section 1. must be an introduction to the paper (Introduction section). In this section, the author could describe shortly the introduction of DNO and show the main contribution of the revision proposed. Try to show if is there some paper with similar proposes.
Response: Thank you for the suggestion. We have chosen to not follow the standard research manuscript sections (e.g. introduction, methods, results, conclusion). As per the author guidelines, it is not necessary to follow this standard structure. We also felt that our structure where we first highlight the importance of diagnostics, and then the importance of diagnostic networks, and why DNO is important was useful in setting up the context to describe the process of DNO and the terms, as per Table 1.
- Figure 1 is interesting and could be better described in order to relate to DNO.
Response: Thank you for this comment. We have added additional information to the notes to the figure: “Example of a national, interconnected diagnostic network: at the higher levels of the network there are fewer sites, higher required levels of expertise and high-throughput diagnostics such as polymerase chain reaction tests for viral detection; at the most basic levels of the network (e.g. clinic and community setting), this includes a large number of sites and rapid diagnostic tests and point of care; all functions are supported by a specimen referral system that ensures timely specimen referral and result delivery for all sites, regardless of service distance.”
- Table 1 indicate and describe the terms of DNO. This table needs to give support using references.
Response: In Table 1, we are trying to define these terms as they relate to DNO within the context of LMICs, and for the vast majority, these terms have not yet been defined in this context. However, where appropriate, we have added references as per the Reviewer’s suggestion.
- The paper, as the title indicates, focuses on low- and middle-income (LMIC) countries, however, the paper does not bring any work or example of a set of countries that the DNO is starting to be used. This is the core of the paper, then needs better an in deep discussion.
Response: We agree with the Reviewer that more detail on DNO examples in LMICs is required. In addition to citing the examples of DNO in 5 LMICS (Kenya, Lesotho, Philippines, South Africa and Zambia) on line 156-161, we have added the following to the Review. “The focus of DNO in Zambia has been on the optimization of viral load access and has seen the number of viral tests performed double within a year[21,22,27,32]. DNO exercises in South Africa have optimized the diagnostic network for CD4 testing, placement of viral load point of care, as well as optimized the specimen routing for multiple diagnostics [26,33,34]. In Lesotho, Kenya and the Philippines, DNO has informed instrument placement and specimen referral within the TB diagnostic network [20,28–30].”
- Also, It is important to bring a short review of how rich countries work with this, such as a short comparative.
Response: Since the focus is on LMICs, we did not feel that it was necessary to provide a review of DNO activities in higher income countries. In the vast majority of higher income countries, DNO is an integral component of their laboratory systems. The focus of this Review is on countries that do not have sophisticated diagnostic networks and are starting the DNO process from a basic network.
- At text "Although a computer model or software is not required for a simple spatial analysis, for our purposes, we are limiting network design and optimization to sophisticated processes that are done with the aid of computer software that enables integration of Geographic Information System (GIS) data, health service capacity and diagnostic demand to prioritize and weigh trade-offs between the various priorities and objectives". The author provides a paper review, but in this sentence, the model is presented as a proposal of the author. Give to clear if is a review paper or a new model proposed.
Response: The authors still believe that a Review paper captures the purpose of this paper which is to “provide concise and precise updates on the latest progress made in a given area of research.” It does not represent original research, nor does this paper propose a new model, rather it provides concise definitions of commonly used terms.
- In general, the paper brings an initial discussion about DNO in LMIC countries
Response: We thank the reviewer for their positive comments.
Reviewer 3 Report
good paper
easily yo understand and well written article
Author Response
Reviewer 3:
- good paper. easily yo understand and well written article
Response: We thank the reviewer for their positive comments.
Round 2
Reviewer 1 Report
The authors addressed all the comments.
Reviewer 2 Report
The authors included the information required in the previous review.
The article has merit and can be published as presented in the final version.